# CRISPR-Cas: ‘The Multipurpose Molecular Tool’ for Gene Therapy and Diagnosis

**DOI:** 10.3390/genes14081542

**Published:** 2023-07-27

**Authors:** Stéphane Sauvagère, Christian Siatka

**Affiliations:** 1Ecole de l’ADN, 19 Grand Rue, 30000 Nîmes, France; 2UPR CHROME, Place Gabriel PERI, 30000 Nîmes, France

**Keywords:** CRISPR-Cas, genetic diseases, gene therapy, diagnosis, molecular tool

## Abstract

Since the discovery of the CRISPR-Cas engineering system in 2012, several approaches for using this innovative molecular tool in therapeutic strategies and even diagnosis have been investigated. The use of this tool requires a global approach to DNA damage processes and repair systems in cells. The diversity in the functions of various Cas proteins allows for the use of this technology in clinical applications and trials. Wide variants of Cas12 and Cas13 are exploited using the collateral effect in many diagnostic applications. Even though this tool is well known, its use still raises real-world ethical and regulatory questions.

## 1. Introduction

Rare diseases raise the question of the medicine of tomorrow and its economic model. These rare diseases affect nearly 30 million people in Europe and more than 350 million worldwide. More than 7000 rare diseases have been identified to date, only approximately 5% of which benefit from approved treatments in Europe. Nearly 75% of patients with rare diseases are children, half of who are under 5 years old. One major objective is to identify and treat them.

Since its discovery in 2010–2012, the CRISPR-Cas system has been widely described. Since the late 1980s, research carried out on bacteria and specifically on immune mechanisms has led to the identification and optimization of a formidable and revolutionary system that can also cut DNA: CRISPR-Cas9. This strange acronym, CRISPR-Cas (clustered regularly interspaced short palindromic repeats/CRISPR-associated), represents a seemingly simple yet formidably effective system for modifying genes to order by generating highly targeted gaps in their DNA. The technical simplicity of its use has ‘burst’ the imagination of genome engineers, though this was previously unthinkable and sometimes frightening. Although the use of this technology, however simple it may be, requires in-depth knowledge of the cell systems used, a perfect command of the genetics of the experimental model, and knowledge of the associated regulations.

The last decades have seen the advent of new scientific technologies such as CRISPR-Cas, which is described as an adaptive immune system that allows bacteria and archaea to defend themselves against viral attacks. The technology derived from these CRISPR-Cas systems, which makes it possible to precisely cleave a genomic sequence, is now the basis of powerful tools for molecular biology and genome editing. The ‘molecular scissors’ in CRISPR-Cas use Cas endonucleases that are programmed and activated with guide RNA to specifically cut a target sequence of RNA or DNA. Some of these Cas enzymes—in particular Cas12a and Cas13a—have, in addition to this cleavage activity directed by an RNA guide, a generic activity of collateral cleavage, in trans, of all nucleic sequences encountered. These different activities of CRISPR-Cas systems have also been exploited to develop promising molecular diagnostic tools. Their applications are numerous, and CRISPR-Cas systems are also used in the context of diagnosis.

The development of genetic and pathogenic tests has improved healthcare outcomes, as they help in diagnosis, monitoring, and updating disease information and reducing negative outcomes. Nucleic acid detection is a key molecular diagnostic procedure that has been steadily increasing in use over the last few decades [1].

The COVID-19 pandemic has challenged the conventional diagnostic field and has revealed the need to create and develop a new generation of point-of-care (POC) solutions [2]. CRISPR-based strategies with the new generation of Cas protein have emerged as promising tools capable of revolutionizing molecular diagnostics.

These systems’ ease of use on a technical level has caused imagination in the genome engineering field to ‘explode’ with previously unthinkable and sometimes frightening ideas. However, the use of this technology, as simple as it is, requires a very thorough knowledge of the cellular systems used, as well as a perfect mastery of the genetics of the experimental model and the associated regulations. Using this tool to perform a modification is not enough; it is necessary for it to be accompanied by structures and complex technological platforms for visualizing, identifying, and selecting the laboratory work obtained with CRISPR-Cas 9. Overall, this coveted modification tool is apparently not yet easy to use in every laboratory on a daily basis. If, in addition, some regulatory limits come to frame its use, it is ultimately a technology that certainly shows promise in all areas of genetics and still has many surprises in store for us in the future.

In this review, we detail the CRISPR-Cas system, its variants, its applications in diagnosis, and strategies for its use in gene therapy.

## 2. DNA Damage Processes and Repair System

DNA is the carrier of genetic information carried by 23 pairs (one paternal copy and one maternal copy) of chromosomes in humans. Its double helix structure, consisting of two complementary strands, each formed of a sequence of nitrogenous bases called nucleotides, is necessary for the conservation and transmission of genetic information. Three successive nucleotides form a codon, which corresponds to a specific amino acid when the corresponding protein is assembled. During divisions, DNA strands are subject to damage such as double-strand breaks (DSBs), which can affect cell viability. After a DSB, a cell can implement two strategies for repairing the break and ensuring its genetic integrity: repair via nonhomologous end joining (NHEJ) and repair via directed homologous recombination (HDR—homology-directed repair) (Figure 1).

Correction via NHEJ most often leads to the suppression of the expression of the repaired gene (generation of a knockout (KO)), because it leads to the random insertion or deletion of nucleotides that are no longer in multiples of three. This type of deletion or insertion, therefore, most often induces a shift in the reading frame of the codons during the translation into protein, leading either to the formation of a truncated protein that is incapable of exercising its main function or the absence of translation into protein.

Conversely, the HDR strategy consists of correcting the break using homologous recombination from an unaltered DNA strand of the sister chromatin (from the other chromosome), which serves as a template strand for the synthesis of repaired DNA [3,4] (Figure 1).

In gene therapy, the HDR strategy appears to be the method of choice, because it allows for the precise correction or even integration of a sequence that is determined and chosen by the experimenter at the level of the gene to be modified. In practice, it is possible to combine molecular scissors with single-strand oligonucleotide DNA (ssODN) in donor oligonucleotides that are homologous to the target sequence to be corrected (except for the part modified by the experimenter), which serves as a template for correction in the homologous recombination induced during HDR [5].

New gene therapy techniques that involve the action of endonucleases are based on the principle of the induction of a DSB at a specific level of DNA, followed by one of the repair strategies listed above. Thus, the endonucleases used in gene therapy always comprise two domains: a recognition and binding domain specific to the sequence of interest and a DNA cutting domain (i.e., the nuclease), which is responsible for creating the DSB. These molecular scissors are, therefore, chimeric proteins capable of recognizing and modifying any sequence of a genome with precision.

## 3. Brief Review of Molecular Tools for Genome Editing before CRISPR

Other systems existed before the discovery of the CRISPR-Cas tool and remain in use. The ability of zinc finger nucleases (ZFNs) to modify genomes was discovered in the mid-1990s. These recombinant proteins are created by fusing a zinc finger protein, which constitutes the recognition domain for a sequence of DNA, with a cleavage domain of a particular endonuclease, *Flavobacterium okeanokoites* endonuclease I (Fok1), capable of cutting DNA [5]. Zinc finger proteins consist of approximately 30 amino acids and have a helix structure. Each ZFN is able to identify and interact with three nucleotides (i.e., a codon). The Fok1 nuclease is associated with six zinc finger protein motifs of the Cys2His2 type to form a chimeric ZFN protein [6]. Since a ZF can only recognize three nucleotides on the target DNA, it is necessary to create constructions consisting of a sequence of several ZFs so that they can recognize a long DNA sequence.

To date, three clinical trials have been set up using the ZFN method; one of them was carried out on T cells isolated from HIV-positive patients [7]. Chimeric ZFN proteins were engineered to modify the expression of CCR5 chemokine receptors, making them resistant to viral infection. These modified cells were then transferred to patients with HIV for therapeutic purposes (NCT02388594) [8].

For several years, the ZFN technique was the only technology that allowed for the creation of specific proteins that bind to a specific region of the genome and allowed for its editing, but more efficient strategies have subsequently emerged.

The TALEN method uses a recombinant protein consisting of a DNA-binding domain, called a transcription activator-like effector (TALE), and the Fok1 endonuclease (the cleavage enzyme). Discovered in 2009, a TALE is a protein naturally secreted by a proteobacterium, named Xanthomonas, which infects a number of plant species, such as rice [8]. In plants, TALEs act by integrating into the nucleus of the host cell and mimicking the action of a transcription factor to modify the gene expression, thus inducing a modification in the plant’s development [9]. This factor consists of a central domain composed of tandem repeats (7 to 30 repeats). These repeats can be chosen and designed in the laboratory so that they can bind to any DNA sequence. Two TALEN proteins bind to opposite strands to allow for the dimerization of Fok1 [5]. Each monomer recognizes a sequence 14 to 20 base pairs (bp) long within a DNA sequence.

A clinical trial for immunotherapy using the TALEN method to treat type B acute lymphoblastic leukemia has been underway in New York since 2018 (NCT04150497) [10,11].

Despite the relative efficiency of molecular scissors of the ZFN and TALEN types, these techniques remain complex, inflexible in terms of the choice of the target DNA sequence, quite expensive, and time consuming to develop. The new CRISPR-Cas9 technique makes it possible to overcome some of these constraints. Its efficiency and, above all, its ease of use make it the perfect choice for genome engineering.

## 4. Structure, Classification, Diversity, Function, and Biotechnological Applications of the CRISPR-Cas Tool

In 2020, Emmanuelle Charpentier (Director of the Max Planck Institute, Berlin, Germany) and Jennifer Doudna (Professor at the University of California, Berkeley, United States) were awarded the Nobel Prize in Chemistry for the discovery of the CRISPR-Cas nuclease system [12]. The history of CRISPR began in 1987, in Japan, when the group of Ishino et al., from the University of Osaka [12], discovered strange repeated sequences of nucleotides (∼21–40 bp) spaced by sequences of ∼20–58 bp, called ‘spacers’ [13,14] (Figure 2). It still took approximately twenty years for Rud Jansen and Francisco Mojica to identify and baptize these CRISPR sequences, in 2002, and five more years for them to be identified as part of an adaptive immune system that allows bacteria to defend themselves against phage infections, as described by Barrangou et al., thus conferring resistance against bacteriophages upon prokaryotes [14,15]. Barrangou et al. presented a kind of adaptive immunity that allowed an infected bacterium to remember previous infections in order to defend itself and better resist a subsequent infection by an identical virus. Indeed, during the first infection by a virus, the viral DNA is integrated into the host genome at the level of the spacers of the CRISPR sequences, and the immunological memory is thus stored in these sites, which is subsequently transcribed into CRISPR RNA (crRNA). During the second infection by a bacteriophage, a process of recognition and defense against it takes place [14]. This process occurs in three stages (Figure 3):

(1) An acquisition phase (or adaptation) during which the new ‘spacer’ sequences that come from the invasive DNA are inserted among the repeated sequences of the CRISPR locus.

(2) A phase of biogenesis that involves the transcription of the CRISPR sequence, as well as that of other genes necessary for the recognition and cleavage of the viral DNA.

(3) An interference phase during which the foreign nucleic acid sequence is targeted and degraded.

### 4.1. Classification Based on the Nature of the Cas Protein

Each endonuclease is composed of an effector domain that allows interaction between the crRNA and the target, a cleavage domain, and a domain involved in the insertion of spacers. Within the adaptive immunity provided by bacterial CRISPR, there are several distinct families depending on the type of Cas protein involved. Cas genes are only found in species with the CRISPR system and are often located near a repeated locus (Figure 2). There is a great diversity of Cas proteins associated with CRISPR; at least 45 different Cas genes are counted today and integrated into a particular classification system [16,17]. The CRISPR system in prokaryotes also includes a great diversity of forms and functions, which have been grouped into two distinct classes. The first is class 1, which contains three subtypes (type I, type III, and type IV). In this group, the effector enzyme is formed from an assembly of several Cas subunits. The second is class 2 (composed of type V, type VI, and type II), which involves only one Cas, which is a large protein capable of coupling to crRNA.

Within class 1, the subunits making up the effector part of the protein vary greatly depending on the type, with several small (ss = small subunit) and large (Cas) subunits playing an effector and cleavage role, respectively. Class 2 is characterized by a single large protein (Cas) acting as an effector and cleavage protein [18] (Figure 2).

Depending on the type of Cas and the subunits involved, the maturation of pre-crRNA into crRNA takes place through different mechanisms. This mechanism within the type II CRISPR-Cas9 system is well known and requires the involvement of the trans-activating CRISPR RNA (tracrRNA) [19]. Because of its relevance, type II has aroused much interest with respect to genome editing and has been the subject of numerous in vitro studies on animal and human cells. CRISPR-Cas 9 is the type of CRISPR system most commonly used in therapeutic research today.

### 4.2. Specificity of the CRISPR-Cas9 Tool

E. Charpentier and J. Doudna, for the first time, demonstrated that the CRISPR-Cas system could be programmed for the cleavage of target DNA in vivo and for the editing of all genomes [18]. This revolutionary discovery allowed them to adapt this technique for any living organism, and this is the main reason they were awarded the Nobel Prize.

This tool is composed of two elements necessary for the modification of the genome: Cas9 enzyme with the function of an endonuclease composed of two cleavage sites, which allows for the formation of a break in the double strands of DNA; guide RNA (gRNA), that condenses the crRNA and tracrRNA into a single RNA and which is chosen to specifically guide Cas9 to the cleavage site. This RNA is composed of a variable sequence of 20 nucleotides complementary to the sequence of the targeted gene and a constant sequence of 42 nucleotides used for attachment to Cas9 [20]. The 20 nucleotides are chosen close to the PAM motif (three nucleotides upstream). In the free state, Cas9 self-inhibits and is, therefore, unable to bind to the target sequence to perform its endonuclease function. Following the binding of the guide RNA to it, Cas9 undergoes a physical rearrangement and, thus, adopts a conformation making it possible to bind to the target DNA [8]. The PAM motif is necessary for the binding of Cas9. Once the Cas9/gRNA complex is bound to the DNA, the RuvC and HNH catalytic sites of Cas9 induce the cleavage of each DNA strand. This triggers a series of events, resulting in the recruitment of the cellular machinery for initiating the DNA repair pathways. Initially, the break is detected by protein complexes which, in turn, recruit enzymes, such as glycosylases, endonucleases, and exonucleases. This multiple complex allows for the nucleotide rearrangement that is necessary for the repairment of the DNA [18]. A DNA polymerase is then recruited to randomly add nucleotides in the case of the NHEJ strategy or to synthesize a DNA strand from a template strand (complementary DNA strand or exogenous ssODN single-strand donor) in the case of the HDR strategy [18,21]. The induction of repair via directed homologous recombination can lead to the invalidation of a gene or the correction of a deleterious mutation. In this context, the exogenously synthesized ssODN donor strand must contain nucleotide sequences homologous to those preceding and following the cleavage site. Between these sequences, the donor strand may contain a sequence of nucleotides that makes it possible to modify the target gene, namely, correcting a mutation, introducing missing exons, decreasing the expression of a gene, or inhibiting it.

### 4.3. In Vivo and In Vitro Applications of the CRISPR-Cas 9 System in Fundamental Research

In view of its considerable advantages, today, the CRISPR-Cas system represents an essential molecular tool for understanding physiological mechanisms, modeling diseases, and conducting therapeutic research. This technique has, indeed, significantly imposed itself; it is now possible to modify a genome at low cost in a few days, which took several weeks or even several months a few years ago using the TALEN and ZFN techniques [22]. Today, researchers are, in fact, capable of generating a considerable number of modifications with precision (such as modifying a gene, correcting a mutation, and inducing a mutation) in cells in culture or in animal models. Depending on the scientific question under study, several types of genetic manipulations can be performed. Genetic invalidation, more commonly called ‘knockout’ (KO), makes it possible to permanently suppress the expression of a gene within a cell type or an organization. It is also possible to reduce the expression of a gene through repression and interference (CRISPRi) or, on the contrary, increase its expression through activation (CRISPRa) without permanently modifying the genome of the cell [20]. Finally, it is possible to specifically modify a genomic sequence to induce or correct a mutation via editing.

The CRISPR-Cas9 system can be introduced into an organism to modify the gene expression of cells in a specific manner using adenovirus vector (AAV)-mediated administration in specific cells [21,22]. Indeed, much research aims to model pathologies using animal models to better understand the mechanisms present in humans. Their purpose is to induce a causal mutation in order to study the physiological consequences linked to it or to correct the causal mutation to gain control over the disease in question [23]. An example of a rare disease of genetic origin that has long been modeled with the CRISPR-Cas9 strategy is Duchenne muscular dystrophy, a syndrome that appears in young children and causes progressive degeneration of all of the muscles of the body. It is mainly caused by mutations in the DMD gene, the longest gene in the body, which is located on the X chromosome; these mutations cause a total loss of dystrophin, a protein necessary for the maintenance of muscle fibers. Several models have been created to study this syndrome, such as a mouse model for which researchers were able to specifically target the two exons of the DMD gene responsible for this myopathy, as well as a nonhuman primate model [24].

Additionally, the discovery of a method for generating induced pluripotent stem cells (iPSCs) in 2006 [25] produced a major asset for many genome-editing approaches. It is now possible to take skin biopsies from patients, put them in culture, and reprogram the fibroblasts obtained into iPS cells using this technique. The editing of the genes of these cells with CRISPR-Cas9 technology makes it possible, on the one hand, to obtain cellular research models for studying the physiopathological mechanisms of genetic diseases, as well as to attempt to correct mutations in the model as a valid therapeutic tool, on the other hand.

## 5. CRISPR-Cas9, Clinical Trials, and Therapeutic Use

CRISPR-Cas9 technology is also opening up new horizons in gene therapy. Indeed, if we manage to detect the gene implicated in a disease, we can, in the long term, eliminate it, correct it, or modify it within the affected organ. Following the success brought by the use of this technique in the scientific world, many clinical trials that have included this technology are currently being set up for curative purposes. Until now, the CRISPR-Cas9 system had never been directly injected into patients. In fact, the clinical studies set up over the past nine years have consisted of using ex vivo therapeutic approaches in which a patient’s cells are transfected/infected in the laboratory and then reinjected into the patient’s tissue. This is particularly the case for cancer research. These studies involve taking cells from patients in order to transform the T cells to make them effective against cancer cells. Patients can then benefit from genetically modified T lymphocytes that recognize tumor cells [26].

To use the CRISPR system, Cas9 and gRNA must be simultaneously expressed within the target cell so that there is a modification of the latter’s genome. Several formats can be used for a CRISPR-Cas9 construct [27]. The CRISPR-Cas9 system can be delivered to a cell in the form of a DNA plasmid, mRNA, or even ribonucleoprotein (RPN), which is a complex combining an mRNA with proteins. These constructs can then be encapsulated in plasmid or viral vectors and delivered to patients. The method involving the DNA plasmid is the simplest and is low cost, but it does not reveal optimal efficiency. In the form of mRNA, the system has the advantage of being very specific, thus limiting the risks of mutagenesis. The introduction of the material in the form of RPN also presents a good specificity of action and has the advantage of being very effective and fast; however, it is seldom used because of its cost [28].

Last March, the CRISPR-Cas9 system was injected into patients for the first time as part of a phase 1/2 therapeutic trial sponsored by Allergan and Editas Medicine (NCT03872479) [28]. This is a nonrandomized, open-label study that includes 18 patients with Leber’s congenital amaurosis, who are adults and children 3 years of age and older and who carry the c.2991 + 1655A > G mutation in intron 26 of CEP290. The trial started in September 2019 to test the drug AGN 151587, also known as EDIT101. This formulation is an AAV5 viral vector containing Cas9 derived from *Staphylococcus aureus* (SaCas9), which is coupled with two guide RNAs that allow for the deletion of the c.2991 + 1655A > G mutation in intron 26 of the CEP290 gene [29]. A study conducted in four American centers (Bascom Palmer Eye Institute Miami, Florida-USA; Massachusetts Eye and Ear Infirmary Boston, Massachusetts-USA; W.K. Kellogg Eye Center, University of Michigan-Arbor, Michigan-USA; Casey Eye Institute, OSHU Portland, Oregon- USA) consisted of a single subretinal injection of AGN 151,587 [29]. Three different doses are being tested first in adults before including children, in whom the intermediate dose and the highest dose will be tested. It is then a question of evaluating the tolerance of the treatment, determining the best-tolerated dose, and studying its effectiveness regarding functional criteria (through tests of mobility, visual acuity, sensitivity to contrasts, color vision, microperimetry, sensitivity retina evaluated with the full-field light sensitivity threshold (FST), pupillary reflex, and the stability of gaze directed at videos) in addition to a structural analysis of macular thickness in OCT and an analysis of quality-of-life questionnaires from treated patients [30].

### First Results of the In Vivo Use of the CRISPR Molecular Chisel to Treat a Rare Genetic Disease

The first results of a phase I clinical trial using the famous gene editor CRISPR directly in the human body were published at the end of June in the *New England Journal of Medicine* [31]. Although it is necessary to wait to know whether the treatment will relieve the symptoms of the six patients treated, the authors are enthusiastic about the first results [32]. CRISPR-Cas9, with its molecular scissors capable of editing genes, for the first time regarding the in vivo treatment of a human, delivered results of the treatment of six individuals with a genetic disease. Data from the phase I clinical trials were published on 26 June 2021, in the *New England Journal of Medicine*. The early findings show that CRISPR seems to work better than current treatments, without serious side effects.

Further human trials involving CRISPR are currently underway, with the tool having shown particular promise in the treatment of numerous conditions, including certain cancers. However, this research is the first published that involves CRISPR’s use directly inside the body [33]; almost all of the other studies performed genome editing in extracted cells, which were then reintroduced. A treatment for restoring sight by injecting CRISPR into a patient’s eye has also emerged, but it has not yet been the subject of any scientific publication.

In 2022, a clinical study for a promising treatment to combat transthyretin amyloidosis began [31]. Here, English and New Zealand teams, in association with the Intellia Therapeutics and Regeneron Pharmaceuticals laboratories, directly injected CRISPR into the blood of people suffering from transthyretin amyloidosis, an inherited disease characterized by attacks on the nervous system, kidneys, and heart. This rare pathology is characterized by a mutation in the TTR gene, which leads to the accumulation of transthyretin proteins in deposits around nerves and organs [31,33].

A gene-editing therapeutic agent, named NTLA-2001, was developed to deactivate this mutation and then given to six patients at two different dosages. After 28 days, a significant reduction in the concentration of the transthyretin protein could be observed: 87% for those who received the highest dosage and 52% for those who received the lowest dosage. Few adverse effects could be detected, and the study stated that those that occurred were mild.

As the study is in phase I, that is, it is only in the phase of verifying the safety profile of the product, it is not the intent to draw conclusions on its effectiveness. However, the authors are enthusiastic about the results of this treatment, as they state in the conclusion of their paper: “The data from the first group of patients in this study provide clinical proof of the concept of in vivo gene editing by CRISPR-Cas9 as a therapeutic strategy”.

Another major advantage of NTLA-2001 [33] is that it acts after a single dose, unlike current drugs which require lifelong intake, achieving an average reduction in the concentration of the transthyretin protein of 81% [34].

The clinical trial is still ongoing, and tests on larger samples are under consideration. In addition, a long-term follow-up of the first six patients is necessary, on the one hand, to assess the action of the treatment on the symptoms of the disease and, on the other hand, to ensure that no off-target gene editing occurred; however, according to the authors of the study, the area where the editing occurred makes this risk ‘low’.

Genome editing therapies offer great hope. Thus, in March 2020, in Portland, Oregon (United States), the very first human therapeutic trial using CRISPR-Cas9 technology began [34]. The technology was used for Leber’s congenital amaurosis (LCA) targeting the specific mutation of the CEP290 gene (ClinicalTrials.gov Identifier: NCT03872479).

Leber’s congenital amaurosis, along with its less severe iterations, such as early-onset retinal dystrophy, is a rare genetic condition that affects the outer retina. It is a mutation in the previously mentioned gene that is the target of the clinical trial (ClinicalTrials.gov Identifier: NCT03872479) started at the University of Oregon [35]. The CEP290 gene codes for nephrocystin 6, a protein important for the development and structure of the primary cilium. This protein, which is located at the connector cilium of the photoreceptors, is essential for maintaining it and for transporting proteins to the external article, among other things [35].

## 6. Applications of CRISPR-Cas Systems in Molecular Diagnostics

### 6.1. A Widely Used System

In molecular diagnosis, the CRISPR-Cas9 system must be combined with other techniques that use nucleic acid amplification (PCR, qPCR, dPCR, and NASBA) for the detection of specific DNA and RNA sequences. With this combination, it is possible to propose ultra-specific biosensing systems for diagnostic purposes [36].

Two types of Cas9-based biosensing systems exist; one of these systems still uses the specific target cleavage activity of the Cas9 effector, and the other uses only the target-specific binding activity, which involves double-mutant Cas9 (D10A, H840A), a nuclease-deficient Cas9 also defined as deathCas9 (dCas9) [37] (Figure 4). By using these possibilities, biosensing strategies can be applied to the genotyping of pathogens and the discrimination of single-nucleotide polymorphisms (SNPs) [38].

In 2016, Pardee et al. combined Cas9 cleavage activity with nucleic acid sequence-based amplification (NASBA) and an isothermal amplification technique based on CRISPR cleavage (NASBACC) [39,40]. They developed a tool with a single-base resolution for identifying the RNA of strains of the Zika virus in vitro [19].

In 2018, Xing et al. developed an original method for the detection of short microRNAs (miRNAs). This method used a CRISPR-Cas9 isothermal exponential amplification reaction (CAS-EXPAR) for site-specific and rapid fluorescent nucleic acid detection [41,42].

Another method based on CRISPR-Cas9 was developed for the detection of drug-resistant pathogens [23] and was applied for the detection of *S. aureus* and *Plasmodium falciparum* [43].

In 2020, Wang et al. developed a kit named CASLFA (CRISPR-Cas9-mediated lateral-flow nucleic acid assay) for the detection of African swine fever virus [44]. The utilization of dCas9, which selectively binds to target DNA but does not cleave it, led to the development of FELUDA (FNCAS9 Editor-Linked Uniform Detection Assay) for molecular testing for SARS-CoV-2 [45].

Using equivalent strategies, Mammoth Bioscience developed the CRISPR-based SARS-CoV-2 DETECTR™, which is a nucleic acid test that overcomes the negative aspects of standard nucleic acid tests, in 2021. This combines nucleic acid amplification with CRISPR-based detection. CRISPR-based detection both enhances its specificity and amplifies signals further. As a result, less time is required for the whole test [46].

### 6.2. Type V CRISPR-Cas Systems: CRISPR-Cas12

One of the main features of Cas12, which has the Cas12a and Cas12b variants [47], is the nonspecific collateral cleavage activity of the Cas12/crRNA/target DNA complex. This complex can cleave any collateral single-stranded DNA (ssDNA) using the active nonspecific catalytic domain of the nuclease. This particular activity has been exploited to create biosensing systems to perform rapid and specific detection of pathogenic DNA samples [48]. The CRISPR-Cas12a complex is activated when it attaches to target dsDNA and cleaves the amplified DNA and FRET probes that emit fluorescence, which is monitored with a fluorometer or, more simply, using paper-based detection techniques. This rapid method is called ‘HOLMES’ because it can be performed in a one-hour, low-cost, multipurpose, and highly efficient system. HOLMES is combined with a DNA endonuclease-targeted CRISPR trans reporter (DETECTR) (Figure 5). These two major CRISPR-Cas12-based diagnostic systems have been applied worldwide [44,46]. HOLMES has been advanced to HOLMESv2, which uses Cas12b instead of Cas12a to detect SNPs and different viruses, such as the Japanese encephalitis virus (JEV) [49]. DETECTR was recently used for the CRISPR-Cas12a-based detection of SARS-CoV-2 [46].

## 7. Another Type of CRISPR-Cas System for Diagnostic Applications: Type VI Cas13

### 7.1. Targeting RNA Using Cas13 and Its Diagnosis Applications

The Cas13 endonuclease was discovered in 2017 by Zhang et al. [50] and has been widely utilized to produce RNA knockdown models, but it has also been applied as a biosensing system for efficient and specific RNA detection (Figure 5).

An important feature of this effector is its collateral RNase cleavage activity, which is triggered after the target sequence is cleaved by the Cas13/crRNA/target RNA complex. Four subtypes of the class 2 type VI Cas13 nuclease have been identified to date. Type VI CRISPR-Cas systems comprise subtypes VI-A, VI-B, VI-C, and VI-D, which is also known as C2c2 or CasRx (type VI-D). They differ in their sizes and sequences, and they all contain two catalytic domains that are responsible for cleaving single-stranded RNA (ssRNA) target sequences [51,52].

Cas13 is a unique RNA-nucleolytic RNA-guided protein that, once activated by binding to the target ssRNA, cleaves nearby RNAs in trans. Cas13 is used in the same way as the previously described HOLMES system; Cas13-based detection systems require isothermal amplification of the target genome, crRNAs, and fluorescent ssRNA probes. The Cas13-based SHERLOCK (specific high-sensitivity enzymatic reporter unlocking) system is the first demonstration of a diagnostic CRISPR-Cas system with indiscriminate cleavage (Figure 5). The ‘heating unextracted diagnostic samples to obliterate nucleases’ (HUDSON) methodology was developed to identify viral genetic material from body fluids, such as urine, blood and its isolates, and saliva, making the SHERLOCK procedure even more efficient [53]. The HUDSON protocol is used for conserved regions within the genetic material of viruses identified using universal-flavivirus recombinase polymerase amplification (RPA) primers [54]. The SHERLOCK and HUDSON protocols can be applied to any virus, but previous testing focused on the diagnosis of flaviviruses, such as the Zika, Dengue, West Nile, and yellow fever viruses [2,19].

Furthermore, Cas13 detection systems have been used in the detection of DNA viruses, such as the Epstein–Barr virus (EBV) [55]. Other recent applications of Cas13 detection systems include the detection of DNA viruses, BK polyomavirus, and cytomegalovirus (CMV) [56].

### 7.2. Even Cas13-Based Systems Can Be Used for Diagnostics for Noninfectious Diseases

Their great applicability in the field of viral diagnostics shows that with this robust system, Cas13-based diagnostics can be implemented for the detection of noninfectious diseases. In the first example, for the detection of graft-versus-host disease in kidney transplants, the system is able to detect abnormal levels of human CXCL9 mRNA, which is an indicator of acute cellular kidney transplant rejection [56]; it is also able to detect specific miRNAs in medulloblastoma patients (miR-19b [57]) and breast cancer cell lines (miR-17 [58]).

One major advantage of the system is its specificity, which permits the discrimination of point mutations (SNPs) and small deletions. This is the case for mutations and deletions in the EGFR and BRAF genes [59], which are commonly used for their specific potential in liquid biopsy sample collection to make cancer diagnosis more affordable and less invasive. Surprisingly, the sensitivity of Cas13-based diagnosis here was similar to that of ddPCR (digital droplet PCR) and qPCR.

## 8. The Use of CRISPR-Cas Systems: Ethical Problems

Until the advent of CRISPR technologies for genome editing, the ZFN and TALEN methods were good alternatives to gene replacement therapies because of their precision and efficiency in cutting and modifying DNA [22]. Nevertheless, they have a number of drawbacks: These two methods require the formation of a protein that is specific to the desired target, which is time consuming and expensive. The ZFN technique does not make it possible to target all of the existing nucleic sequences, and the construction of the chimeric protein that allows for the cleavage cannot be carried out by a researcher and requires a long process performed by external service providers. TALEN, for its part, is very large (3 kb), which makes its encapsulation within a transfection vector very complex. This aspect limits its application in therapy in comparison with ZFN (1 kb) [60]. Conversely, the CRISPR-Cas9 strategy has the advantage of being able to be designed to recognize any DNA sequence in the genome because of the simplified design of guide RNAs. A second advantage of CRISPR-Cas9 lies in the possibility of constructing a multiplex so that the system can recognize and modify several sites at the same time. A single application could then simultaneously allow for the modification of several genes in the same cell [24] in Cas13-based diagnosis, which is similar to that with ddPCR (digital droplet PCR) and qPCR.

### Problems Encountered When Using CRISPR-Cas9

Despite the rise of CRISPR-Cas9 genetic engineering around the world, several problems can be encountered when using it. Several ethical considerations notably limit its use for gene therapy. Indeed, the Oviedo Convention, implemented in 1999 by the Member countries of the European Council, prohibits any genetic modification that would be transmissible to descendants; thus, precautions must be taken to avoid this during clinical interventions [35]. Other nonmember states of the European Council, such as Canada, the United States of America, Japan, and the Holy See, participated in the development of this convention. The use of CRISPR-Cas9 on embryos—in particular, to correct genetic defects transmitted by the parents, which can lead to serious pathologies—is supervised in both fundamental and clinical research to avoid the risk of eugenics. In China, a clinical trial started in 2018 and, led by He Jiankui, was conducted in order to modify the C-C chemokine receptor type 5 (CCR5) gene in human embryos [61]. Regulations at the international level would, therefore, be necessary to ensure an ethical limit to this practice around the world. Nevertheless, the use of CRISPR-Cas9 on somatic cells, such as in the clinical trial NCT03872479, are authorized because they do not modify germ cells and are surrounded by methodological precautions [62,63]. In addition to the ethical and technical problems, the risks associated with the occurrence of unexpected mutations in the genomes of patients must be considered. Indeed, the use of CRISPR-Cas9 in any cell can lead to nonspecific modifications at off-target sites in the recipient genome. Despite the specificity with which gRNA is constructed and the very careful choice of PAM sites, parts of the genome are likely to have similarities with the targeted part(s). This may lead to the creation of undesirable mutations in patients. In order to reduce this problem that is faced when using CRISPR-Cas 9, innovations and methods for predicting and limiting the occurrence of off-target effects are needed to facilitate the clinical application of this technology. In this sense, to overcome these problems, new molecular techniques inspired by CRISPR-Cas9 are the subject of research. David Liu’s team at the Massachusetts Institute of Technology and Harvard University has been developing strategies for modifying the genome ‘base by base’, which is known as ‘base editing’ and ‘prime editing’, since 2017 [22,64].

## 9. Conclusions

A revolutionary genetic tool for modifying DNA in living beings, CRISPR-Cas9 is seen as the cause of a real boom in the world of genetic engineering. Because of its accessibility and ease of use in both basic research and medicine, it remains, by far, the most widely applied technique currently used to modify DNA.

In recent years, great innovations have emerged in the use of each of the CRISPR-based nucleic acid detection systems described above. CRISPR-Cas systems must be combined with another method for pre-amplifying nucleic acids. Combining isothermal amplification with Cas13-based detection enables very sensitive and specific detection of cell-free DNA from liquid biopsy samples, which makes it more affordable and obviates the need for expensive laboratory equipment. These platforms are inexpensive, simple, and do not require the use of special instrumentation, suggesting that they could democratize access to disease diagnostics.

In human gene therapy, despite the ethical and technical problems that may be encountered with its use, this technology opens many doors for a new generation of gene therapy for treating hereditary genetic diseases that have remained, for the most part, incurable. The first clinical trial, which started at the University of Oregon to treat ACL, brings a glimmer of hope for the therapy of hereditary retinal diseases and is only the start of a considerable therapeutic evolution.

At this time (June 2023), on ‘clinicaltrial.gov’, one can observe that 47 studies can be found for CRISPR-Cas, compared to November 2022, when there were 39, and in March 2023, there were 43. We are observing a rise in clinical trials using CRISPR-Cas, a progression that will undoubtedly increase in the coming years. All of these elements are linked to the increasingly fine mastery of the CRISPR-Cas tool.

## Figures and Tables

**Figure 1 genes-14-01542-f001:**
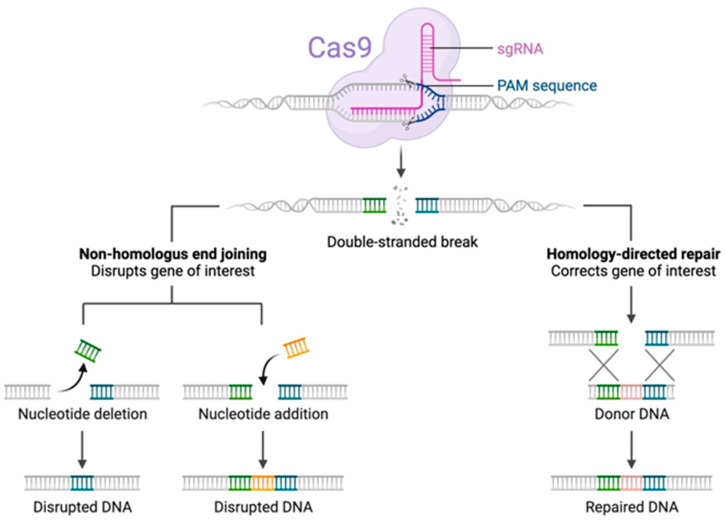
NHEJ and HDR cell reparation systems (credit: Esmée Dragt (creator) and Louis Ngai (Biorender)). CRISPR-Cas9 is a powerful tool for genome engineering. Together with an sgRNA, the Cas9 complex recognizes a specific sequence—the protospacer. This is only possible if this sequence is followed by a protospacer-adjacent motif (PAM). When Cas9 binds, a dsDNA break is generated. Then, nonhomologous end joining or homology-directed repair can occur, leading to mutations or gene changes, respectively. (Colors used in NHEJ represent different sequences, colors for HDR represent homologous recombining sequences).

**Figure 2 genes-14-01542-f002:**
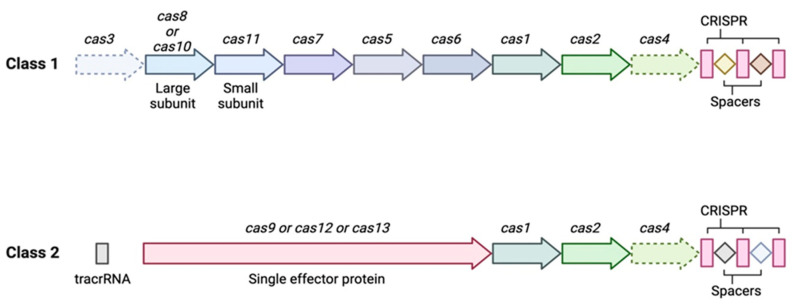
Generic organization of class 1 and class 2 CRISPR-Cas loci. The classification of class 1 CRISPR-Cas systems, which include types I, III, and IV, has remained relatively stable since the last version was described in 2015. The scheme for this classification in 2015 included 12 subtypes. The update of this classification distinguished four new subtypes: III-E, III-F, IV-B, and IV-C. Moreover, experimental data allowed for the observation of the incorporation of new spacers by the systems of the I-U subtype, resulting in reclassification of this subtype into subtype I-G [16,17]. Class 2 CRISPR-Cas systems include types II, V, and VI. Because of the search for new potential genome editing tools, this class has undergone drastic expansion since 2015. From the two types and four subtypes in 2015, class 2 has expanded to 3 types and 17 subtypes. The major characteristic of these types is the presence of a large multidomain protein necessary for interference, such as Cas9. New findings include multiple variants of the V-type and VI-type systems, including V types that cleave RNA.

**Figure 3 genes-14-01542-f003:**
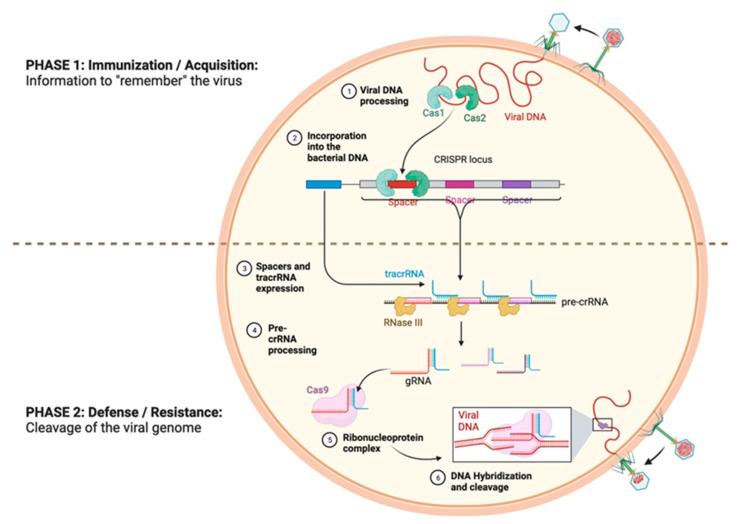
CRISPR-Cas system (credit: Sebastián Felipe González Moraga (creator) and Nima Vaezzadeh; Biorender). The CRISPR-Cas9 system confers bacteria with adaptive immunity against invading bacteriophages. Following infection, Cas1 and Cas2 incorporate a short segment of the viral genome as spacers within the CRISPR locus of the bacterial genome. Upon reinfection with the same bacteriophage, the CRISPR locus is expressed as pre-crRNA, along with tracrRNA. This pre-crRNA is processed to yield guide RNAs (gRNAs) that bind the ribonucleoprotein Cas9, as well as complementary segments of the invading viral genome, prompting its cleavage by Cas9.( Dotted line represent the distinction between phase 1 and Phase 2).

**Figure 4 genes-14-01542-f004:**
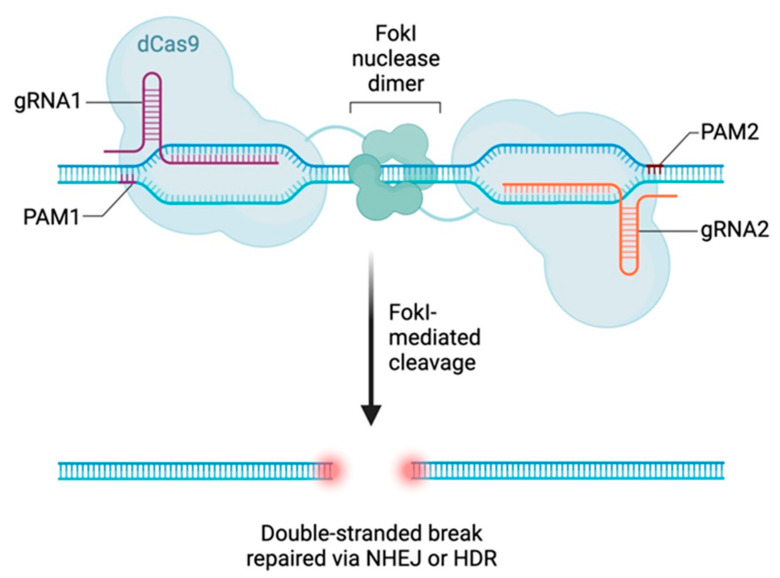
The dCas FokI system (https://www.addgene.org/crispr/dcas9foki/, accessed on 16 July 2023). ‘CRISPR Technology: Gene Editing dCas9-FokI’ is an improved CRISPR-Cas9 system—the FokI-dCas9 system—for precision medicine and, in particular, targeting phenylketonuria (PKU) and other monogenic metabolic diseases. The FokI-dCas9 system can greatly improve the specificity of genome editing. In contrast to the standard system, it requires dimerization of the FokI-dCas9-sgRNA complex, meaning that monomeric FokI-dCas9-sgRNA is unable to cut DNA strands, thus substantially reducing the chances of contaminating off-target effects.

**Figure 5 genes-14-01542-f005:**
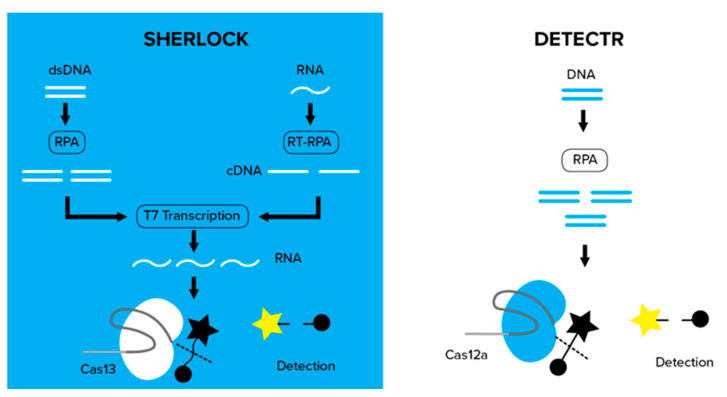
SHERLOCK and DETECTR (https://blog.addgene.org/finding-nucleic-acids-with-sherlock-and-detectr, accessed on 16 July 2023). Both SHERLOCK and DETECTR harness the promiscuous cleavage and degradation of neighboring ssRNA and ssDNA with Cas13 and Cas12a, respectively, to cleave and activate a reporter. The detectable signal from this reporter can be measured and quantified to determine the presence and quantity of DNA, RNA, or a mutation of interest. Together, SHERLOCK and DETECTR demonstrate the power of CRISPR-based diagnostics (Star represent fluorophore, yellow when the fluorophore is visible and black when it is quenched by the quencher represented by the black circle). (credit: Alyssa Cecchetelli).

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
