# Peer review of "CRISPR-Cas: ‘The Multipurpose Molecular Tool’ for Gene Therapy and Diagnosis"

_genes, 2023, doi:10.3390/genes14081542_

Round 1
Reviewer 1 Report
The authors summarized the DNA damage process, repair system, and molecular tools (ZFN, TALEN and CRISPR) in genome editing. Meanwhile, they reviewed the classification of the Cas, and the functions and applications of Cas9. The authors further outlined the applications of CRISPR-Cas system in gene therapy and diagnosis. Lastly, the ethical problems and challenges existed so far were pionted out during in the use of CRISPR-Cas system.
However, the paper have several problems as following:
1) In this manuscript, the Cas12 and Cas13 were considered as a variant of Cas9 (in the line 462 and line 477), which is not appropriate.
2) It would be better if Figure 4 showed the two types of Cas9-based biosensing systems, instead of dCas9.
3) The format of the references needed to be checked.
4) There are many misspelled words, such as DBS (line 67), overwiew (line 150), pre-CRNA (line 236), and so on.
Need extensive revision on English writing.
Author Response
Dear Reviewer,
The whole team would like to thank you for your comments and suggestion concerning the reviewing of our article. The extensive English revision will be performed by a native English speaker.
We provide a point-by-point response concerning your comments.
1) In this manuscript, the Cas12 and Cas13 were considered as a variant of Cas9 (in the line 462 and line 477), which is not appropriate.
Response 1 :
In many Articles Cas12 and Cas13 are presented as variant but, it true that the variant are mainly the mutant Cas9, so it is important to precise that these are different Classes. For these reasons we change the titles of the paragraphs:
“6.2 Type V CRISPR-Cas systems: the CRISPR-Cas12”
“7. Another type of CRISPR-Cas system for diagnostic applications: Type VI CRISPR-Cas13”
2) It would be better if Figure 4 showed the two types of Cas9-based biosensing systems, instead of dCas9.
Response 2 :
We chose to present the dCas9 system, because its application is strategic for gene regulation and diagnosis.
Considering your suggestion, we include an additional figure 5 describing SHERLOCK and DETECTR.
3) The format of the references needed to be checked.
Response 3 :
The format of the references will be checked with the editor, to fit with the standards of the journal.
4) There are many misspelled words, such as DBS (line 67), overview (line 150), pre-CRNA (line 236), and so on.
Response 4 :
The correction on the misspelled word are made, the article is submit to a native English Speaker expert in field for a complete review.
Thank you for your consideration of our responses to your detailed review.

Reviewer 2 Report
Overall Summary:
Sauvagère and Siatka have written a brief review of the CRISPR-Cas systems that are being utilized in the scientific community, and briefly describing its use as a therapeutic or diagnostic tool.
While this article does a good job incorporating the various aspects of this technology, I believe that this sort of review has been written several times over since the discovery of CRISPR-Cas and does not bring to light any new information. Further, the article is laden with grammatical and typographical errors, such that if this article were to move forward with publication, it should be read and edited by a professional scientific editor prior to resubmission. There is specific nomenclature and structure used in scientific writing that I feel are lacking throughout this manuscript that need to be addressed.
Additionally, I feel that the front end of the manuscript is full of detail and explanation, but towards the end becomes sparse and not as thorough. If the manuscript is to be resubmitted, I would suggest reducing the front-end explanations of CRISPR and elaborate on its new uses in diagnostics and how it has been adapted in recent years. Specifically, elaborate on base-editing and prime-editing, both of which are topics there are much fewer reviews on, but still based on the CRISPR technology.
Suggested Modifications:
Suggested title change: Remove the word “even”, or modify the entire title to remove this word.
There isn’t a single explanation of what CRISPR stands for in the entire manuscript, which is the entire topic of discussion.
Line 35: “very numerous” is somewhat redundant. Suggest a change to just “numerous”.
Line 44: “…have emerged promising tools…”. Suggest a change to: “…have emerged as promising tools”.
Line 52: CRISPR/Cas 9 is used, where it has been previously listed as CRISPR-Cas. Suggestion is to stay consistent with either forward slash or hyphen.
Line 57 sentence is somewhat confusing grammatically. Suggested change: “In this review, we will detail the CRISPR-Cas system, its variants, as well as its applications in diagnosis and strategies of use in gene therapy.”
Line 82: a small suggestion. Change the sentence to read “… combine the molecular scissors with single strand oligonucleotide DNA (ssODN) donor..”.
Line 84: remove the word “and”.
Line 94: same suggestion as above, stay consistent with nomenclature used, specifically when choosing CRISPR/Cas versus CRISPR-Cas. My suggestion is to look through the entire manuscript and make these changes.
Line 94: “Before the discovery of the CRISPR/Cas tools, other systems were…”
Line 95 and throughout the manuscript, I would suggest introducing the full name of an acronym before using the acronym. There are multiple instances of the reverse occurring, such as here with “ZFNs”.
Line 122: Fok1
Line 150: misspelling and grammar changes. “Structure, Classification, Diversity, Function, and Biotechnological Applications of the CRISPR-Cas Tool”.
Line 152: “In 2020, Jennifer Doudna and Emmanuelle Charpenteir were awarded the Nobel Prize in Chemistry for the discovery of the CRISPR-Cas nuclease system.”
Line 154: grammatical change. Remove “is interested” and change to “…from the University of Osaka discovered strange repeated …”.
Line 160: what is the pronoun “They” describing?Line 167: Is the acquisition phase #1? Or is the 1) in line 168 the first? Please clarify. The way it is written, it suggests the acquisition phase is 1).
Line 192: …”experimental data allowed to observe”. Grammatically, I believe this should read, “…experimental data allowed for the observation of incorporation of new spacers by the systems of subtype I-U, resulting in reclassification of this subtype into subtype I-G (Source)”.
Sentence starting Line 195: Standard writing practice suggests spelling out numbers ten and under and using numerals for larger numbers above ten.
Line 207: The common way to write this is trans-activating CRISPR RNA (tracrRNA).
Line 237: Comma after Cas9.
Line 243: Commas after have and time.
The paragraph starting on Line 241: This is redundant with a previous paragraph describing their Nobel Prize.
Lines 250-251: “..as well as a guide RNA (gRNA), which condenses the crRNA and tracrRNA into a single RNA, chosen to…”
Line 261: This will trigger a series of events, resulting in the recruitment of the cellular machinery to initiate the DNA repair pathways”.
Sentence starting on Line 262: Run-on sentence. Consider breaking it into multiple sentences.
Line 292: What do you mean by “some of its cells”? Yes, the system can be introduced into organisms to modify gene expression, but without elaborating on how this is accomplished (i.e., AAV-mediated administration to target hepatocytes), I think this statement should be removed.
Sentence starting on Line 298: This sentence starts by sounding like there are only two mutations that cause DMD. I believe the authors are suggesting that this disorder is caused by mutations in the DMD gene, resulting in the loss of dystrophin. Also, using the word mainly suggests there are alternate ways for DMD to occur outside of mutations in the DMD gene.
Line 302: A mouse rat model? These are two separate model organisms. Also, a citation for these studies/models would be beneficial.
Line 305: Remove “on the other hand”. Start the paragraph with “Additionally, the discovery of how to generate induced pluripotent stem cells…”. And then remove the next sentence and use the citation on the first sentence.
Line 315: Capitalize “Therapeutic”. Also, stay consistent with grammatical choice on headings, especially capitalization choice. This will be dictated by the editor/journal.
Line 394: Replace harmlessness with “verifying the safety profile..”.
Line 398: Include the source again.
Line 407: Confusing first sentence. Rewrite.
Line 410: Remove “northwestern”.
Line 410 and others: Common nomenclature for listing clinical trial information is to denote it as (ClinicalTrials.gov Identifier: NCT03872479).
Line 422: “The System CRISPR-Cas9 with his sensitivity”…? This sentence needs to be rewritten.
Line 432: “In 2016, Pardee et al.”.
Line 434: “they develop a tool…” Grammar. Rewrite.
Line 436, 443 and others: Use et al, not “& col” or “& al”.
Line 463: Again, “his”? Do the authors mean “the”?
Line 470 sentence: Rewrite.
Paragraph 501: The authors spell out Epstein-Barr virus twice and use the acronym in both instances. It only needs to be introduced once and all subsequent iterations can use the acronym.
Line 536: “… allow FOR the modification”.
Line 539: Cas9, not Cas 9.
Line 548: Remove “very”.
Line 553: Comma after NCT03872479
Line 575 sentence: Grammar errors. Rewrite.
As stated in the review, I believe this manuscript should be reviewed and edited by an editor with experience in scientific writing.
Author Response
Reference : Response to Reviewer 2 - Comments
Dear Reviewer,
The whole team would like to thank you for your detailed, technical comments and precise suggestion concerning the reviewing of our article. We consider your comment as major suggestions to improve the quality of our article.
We provide a point-by-point response concerning your comments.
- Suggested title change: Remove the word “even”, or modify the entire title to remove this word.
Response : Following your comment we change the title insisting in the multipurpose: “CRISPR-Cas: “The Multipurpose Molecular Tool” for gene therapy and diagnosis”
- There isn’t a single explanation of what CRISPR stands for in the entire manuscript, which is the entire topic of discussion.
Response : we add this paragraph “Since its discovery in 2010-2012, the CRISPR-Cas system has been widely described. Since the late 1980s, research carried out on bacteria, specifically on immune mechanisms, has led to the identification and optimisation of a formidable revolutionary system that can also cut DNA: CRISPR-Cas9. This strange acronym CRISPR-Cas (Clustered Regularly Interspaced Short Palindromic Repeats/Crispr Associated) represents a seemingly simple yet formidably effective system for modifying genes to order by generating highly targeted gaps in the DNA. The technical simplicity of its use has 'exploded' the imagi-nation of genome engineering, which was previously unthinkable and sometimes frightening. However, the use of this technology, however simple it may be, requires in-depth knowledge of the cell systems used, a perfect command of the genetics of the experimental model and the associated regulations.”
- Line 35: “very numerous” is somewhat redundant. Suggest a change to just “numerous”.
Response : The change is made
- Line 44: “…have emerged promising tools…”. Suggest a change to: “…have emerged as promising
tools”.
Response : The change is made
- Line 52: CRISPR/Cas 9 is used, where it has been previously listed as CRISPR-Cas. Suggestion is to stay consistent with either forward slash or hyphen.
Response : the correction with CRISPR-Cas is integrated
- Line 57 sentence is somewhat confusing grammatically. Suggested change: “In this review, we will detail the CRISPR-Cas system, its variants, as well as its applications in diagnosis and strategies of use in gene therapy.”
Response : the new sentence is integrated
- Line 82: a small suggestion. Change the sentence to read “… combine the molecular scissors with single strand oligonucleotide DNA (ssODN) donor..”.
Response : This excellent suggestion is integrated in the text
- Line 84: remove the word “and”.
Response : The change is made
- Line 94: same suggestion as above, stay consistent with nomenclature used, specifically when choosing CRISPR/Cas versus CRISPR-Cas. My suggestion is to look through the entire manuscript and make these changes.
Response : The change is made, and all the modifications have been performed in the entire manuscript using CRISPR-Cas
- Line 94: “Before the discovery of the CRISPR/Cas tools, other systems were…”
Response : “advent” his change by “discovery”
- Line 95 and throughout the manuscript, I would suggest introducing the full name of an acronym before using the acronym. There are multiple instances of the reverse occurring, such as here with “ZFNs”.
Response : The change is made
- Line 122: Fok1
Response : the correction is made
- Line 150: misspelling and grammar changes. “Structure, Classification, Diversity, Function, and Biotechnological Applications of the CRISPR-Cas Tool”.
Response : this proposition is integrated
- Line 152: “In 2020, Jennifer Doudna and Emmanuelle Charpenteir were awarded the Nobel Prize in Chemistry for the discovery of the CRISPR-Cas nuclease system.”
Response : this proposition is integrated
- Line 154: grammatical change. Remove “is interested” and change to “…from the University of Osaka discovered strange repeated …”.
Response : the change is made
- Line 160: what is the pronoun “They” describing?
Response : “They” is modified by “Barrangou et al”
- Line 167: Is the acquisition phase #1? Or is the 1) in line 168 the first? Please clarify. The way it is written, it suggests the acquisition phase is 1).
Response : In the text the clarification of each phase is made using tree different points 1), 2), 3).
- Line 192: …”experimental data allowed to observe”. Grammatically, I believe this should read, “…experimental data allowed for the observation of incorporation of new spacers by the systems of subtype I-U, resulting in reclassification of this subtype into subtype I-G (Source)”.
Response : the change is made as suggested and completed by the references 16 and 17
- Sentence starting Line 195: Standard writing practice suggests spelling out numbers ten and under and using numerals for larger numbers above ten.
Response : the change is made: “From two types and four subtypes in 2015, class 2 has expanded to tree types and 17 subtypes”
- Line 207: The common way to write this is trans-activating CRISPR RNA (tracrRNA).
Response : the change is made
- Line 237: Comma after Cas9.
Response : the Comma is added
- Line 243: Commas after have and time.
Response : the two Commas are added
- Paragraph starting on Line 241: This is redundant with a previous paragraph describing their Nobel Prize.
Response : To be more precise yhis part is modified as “E. Charpentier and J. Doudna have, for the first time, demonstrated that the system CRISPR-Cas could be programmed for the cleavage of a target DNA in vivo, and for the editing of all genomes [18]. This revolutionary discovery allowed them to develop this technique for any living organism, and this is the main reason why awarded the Nobel Prize.”
- Lines 250-251: “..as well as a guide RNA (gRNA), which condenses the crRNA and tracrRNA into a single RNA, chosen to…”
Response : the modification is made as suggested
- Line 261: This will trigger a series of events, resulting in the recruitment of the cellular machinery to initiate the DNA repair pathways”.
Response : the modification is made as suggested
- Sentence starting on Line 262: Run-on sentence. Consider breaking it into multiple sentences.
Response :This Run-on sentence is rewritten as
“Initially, the break will be detected by protein complexes which will in turn recruit enzymes as glycosylases, endonucleases and exonucleases. This multiple complex allows a nucleotide rearrangement necessary for repairing the DNA [18]. A DNA polymerase will then be recruited to add nucleotides randomly in the case of the NHEJ strategy or to synthesize a DNA strand from a template strand (complementary DNA strand or ex-ogenous ssODN single-strand donor) in the case of the HDR strategy [18], [21]”
- Line 292: What do you mean by “some of its cells”? Yes, the system can be introduced into organisms to modify gene expression, but without elaborating on how this is accomplished (i.e., AAV-mediated administration to target hepatocytes), I think this statement should be removed.
Response : We precise es follow “The CRISPR-Cas9 system can be introduced into an organism to modify the gene expression of cells and in a specific way using Adeno Viruses Vector (AAV) mediated administration in specific cells [21] [22].”
- Sentence starting on Line 298: This sentence starts by sounding like there are only two mutations that cause DMD. I believe the authors are suggesting that this disorder is caused by mutations in the DMD gene, resulting in the loss of dystrophin. Also, using the word mainly suggests there are alternate ways for DMD to occur outside of mutations in the DMD gene.
Response : the “two mutations” are deleted
- Line 302: A mouse rat model? These are two separate model organisms. Also, a citation for these studies/models would be beneficial.
Response : this is a mousse model described by the ref 24
- Line 305: Remove “on the other hand”. Start the paragraph with “Additionally, the discovery of how to generate induced pluripotent stem cells…”. And then remove the next sentence and use the citation on the first sentence.
Response: The correction follws instruction and the new version id: “Additionally, the discovery of how to generate induced pluripotent stem cells (iPSCs) in 2006 [25] is a major asset for many genome editing approaches. It is now possible to take skin biopsies from patients, put them in culture, and reprogram the fibroblasts obtained into iPS cells by this technique”.
- Line 315: Capitalize “Therapeutic”. Also, stay consistent with grammatical choice on headings, especially capitalization choice. This will be dictated by the editor/journal.
Response : the correction is made
- Line 394: Replace harmlessness with “verifying the safety profile..”.
Response : the correction is made
- Line 398: Include the source again.
Response : the source 33 is included
- Line 407: Confusing first sentence. Rewrite.
Response : the sentence is corrected as follows “Genome editing therapies offer great hope. So, in March 2020 in Portland, Oregon (United States) the very first human therapeutic trial using CRISPR-Cas9 technology was used [34]. It was on Leber’s Congenital Amaurosis (LCA) targeting the specific mutation of the CEP290 gene (NCT03872479, https://clinicaltrials.gov/)”.
- Line 410: Remove “northwestern”.
Response : integrated in the previous response
- Line 410 and others: Common nomenclature for listing clinical trial information is to denote it as (ClinicalTrials.gov Identifier: NCT03872479).
Response: we adapt the correction in the text as suggested
- Line 422: “The System CRISPR-Cas9 with his sensitivity”…? This sentence needs to be rewritten.
Response : the sentence is corrected as follows “In molecular diagnosis, the CRISPR-Cas9 system must be combined with other techniques, using nucleic acid amplification (PCR, qPCR, dPCR, NASBA), for the de-tection of specific DNA and RNA sequences.”
- Line 432: “In 2016, Pardee et al.”.
Response: the correction is made
- Line 434: “they develop a tool…” Grammar. Rewrite.
Response: the correction is made “They have developed”
- Line 436, 443 and others: Use et al, not “& col” or “& al”.
Response: in the whole text all is corrected as “et al.”
- Line 463: Again, “his”? Do the authors mean “the”?
Response: yes, this is the mean “it is changing.”
- Line 470 sentence: Rewrite.
Response: the run on sentence is corrected as follows : “This rapid method is called ‘HOLMES’ because it can be done in One-HOur Low-cost Multipurpose highly Efficient System. HOLMES is combined with DNA Endonuclease Targeted CRISPR Trans Reporter (DETECTR). These two major CRISPR-Cas12-based diagnostic systems, have been applied worldwide [44], [46] HOLMES advanced to HOLMESv2, which uses Cas12b instead of Cas12a to detect SNPs and different viruses, such as the Japanese encephalitis virus (JEV) [49]. DETECTR has been used recently for the CRISPR-Cas12a-based detection of SARS-CoV-2 [46].”
- Paragraph 501: The authors spell out Epstein-Barr virus twice and use the acronym in both instances. It only needs to be introduced once and all subsequent iterations can use the acronym.
Response: We integrated this remark and erase the last citation of “Epstein-Barr virus”. “Furthermore, the Cas13-detection system has been used in DNA virus detection, such as the Epstein–Barr virus (EBV) [55]. Other recent application of the Cas13-detection system has been used for DNA virus detection, BK polyomavirus and cytomegalovirus (CMV) [56].
- Line 536: “… allow FOR the modification”.
Response: the correction is made
- Line 539: Cas9, not Cas 9.
Response: the correction is made
- Line 548: Remove “very”.
Response: the correction is made
- Line 553: Comma after NCT03872479
Response: the comma is set
- Line 575 sentence: Grammar errors. Rewrite.
Response: the correction is made as follows: “In recent years a great innovation is emerging using each of the CRISPR-based nucleic acid detection systems described above, the CRISPR-Cas system must be combined with another method for preamplifying nucleic acids”.
Thank you for your consideration of our responses to your detailed review.
